# Lipids in Archaeological Pottery: A Review on Their Sampling and Extraction Techniques

**DOI:** 10.3390/molecules27113451

**Published:** 2022-05-26

**Authors:** Anna Irto, Giuseppe Micalizzi, Clemente Bretti, Valentina Chiaia, Luigi Mondello, Paola Cardiano

**Affiliations:** 1Department of Chemical, Biological, Pharmaceutical and Environmental Sciences, University of Messina, 98168 Messina, Italy; clemente.bretti@unime.it (C.B.); valentina.chiaia@studenti.unime.it (V.C.); lmondello@unime.it (L.M.); paola.cardiano@unime.it (P.C.); 2Chromaleont s.r.l., c/o Department of Chemical, Biological, Pharmaceutical and Environmental Sciences, University of Messina, 98168 Messina, Italy; 3Unit of Food Science and Nutrition, Department of Medicine, University Campus Bio-Medico of Rome, 00128 Rome, Italy

**Keywords:** lipids in pottery, archaeological biomarkers, ancient pottery, ageing study, sampling of lipids, lipid extraction, lipid derivatization

## Abstract

Several studies have been performed so far for the effective recovery, detection and quantification of specific compounds and their degradation products in archaeological materials. According to the literature, lipid molecules are the most durable and widespread biomarkers in ancient pottery. Artificial ageing studies to simulate lipid alterations over time have been reported. In this review, specific lipid archaeological biomarkers and well-established sampling and extraction methodologies are discussed. Although suitable analytical techniques have unraveled archaeological questions, some issues remain open such as the need to introduce innovative and miniaturized protocols to avoid extractions with organic solvents, which are often laborious and non-environmentally friendly.

## 1. Introduction

Organic residues in archaeology refer to a wide variety of amorphous organic remains commonly associated with ceramic containers or tools, found in archaeological contexts. The extraction and analysis of such organic residues from pottery can provide answers to a variety of archaeological questions about diet, food storage and processing, rituals and medical practices, trade and the use of commodities, domestication of animals, etc., thus contributing to unveiling crucial hints about daily life of the ancient societies. In fact, it is well known that ceramics were used, not only for decorative purposes, but, significantly, for a wide variety of functions [1,2]. Broadly speaking, culinary pottery can be distinguished according to its use, i.e., storage containers, processing vessels (employed for grinding, crushing, mixing, marinating, boiling, roasting etc.) and tableware (for eating or serving) [3,4,5]. Organic residues can be found both in the inner and outer part of the pottery container as visible remains, surface deposits and encrustations. A surface residue on the vessel’s outer walls may derive from soot deposited during cooking activities on the fire. In contrast, visible burnt residues adhering to the inner wall of a container can result from the charring of food [6,7,8,9]. More commonly, organic residues occur as invisible absorbed material within the porous unglazed vessel wall. Both visible and invisible organic residues can derive not only from the processing or storage of foodstuffs, but also from nonculinary practices, i.e., sealing or waterproofing purposes [8,10], to create coatings of the inner surface of ceramic [11]. However, containers could be used for multiple functions and/or reused or recycled for different purposes over time. From the aforementioned, it is clear that establishing specific functions and uses of pottery vessels is a truly daunting task. Consequently, the most innovative technologies in analytical chemistry must be closely merged with adequate archaeological guidance in order to unravel the origin of organic residues and gain crucial hints on ancient human activities.

To date, lipids (i.e., fats, waxes and resins) represent one of the main chemical classes of substances investigated in archaeological pottery. This is due to their hydrophobicity that makes them less prone to loss by solubilization than other more soluble organic compounds (i.e., carbohydrates, proteins) [8,12], thus limiting their percolation and allowing their persistence in the original site. Conversely, more polar substances are more susceptible to decay, especially those containing nitrogen and phosphorous atoms [13]. Although the resistance to decay and the hydrophobic character of the lipids could make them excellent candidates as “archaeological biomarkers” [14,15,16,17], it should be highlighted that most of them are featured by reactive functional groups that fatally lead to their decay over time. In addition, the strict relationship between edible lipid substances and preservation should be kept in mind. The main characteristic of foodstuffs is their digestibility in the gastrointestinal tract. This means that lipid molecules are also likely susceptible to degradation by microorganisms in the burial environment [18]. Consequently, the lipids may undergo in situ chemical or microbiological degradations over time. This fact further complicates the interpretations on the origin of lipid matter [3].

It is well established, however, that some archaeological environments can retard the degradation of lipids, for instance, very dry climactic conditions [19,20,21] or acidic soils [4,22,23] may retard their decay. In addition, the entrapment of lipid molecules within a ceramic matrix preserves or, at least, retards their alteration [13,24]. The degree of preservation is highly dependent on the chemical and physical conditions (pH, temperature, biomass and humidity) of the burial environment. This means that preservation of the lipid matter in an archaeological context depends mainly on the presence of favorable conditions. The literature data indicate that pottery vessels possess suitable characteristics for absorbing organic material and preserving it during burial over millennia [25,26,27], whilst the contamination of organic residues from the burial soil occurs only rarely. The entrapment of lipids in organic or mineral matrices generally limits their loss by microbiological degradation. In fact, the access of exocellular enzymes produced by degrading microbes to lipid matter would be prevented, especially in highly dense or vitrified materials due to their low porosity and permeability [26]. Lipids are also well-preserved in carbonized organic residues on pottery [28], probably due to microencapsulation that inhibits microbial activities. The encapsulation of organic residues within clay surfaces may also limit the access of microorganisms, but the presence of water and other reactive species may cause some chemical degradation processes, such as hydrolysis or oxidation, leading to the formation of specific “archaeological biomarkers” [26]. For example, the partial hydrolysis of triacylglycerols (TAGs) leads to the formation of diacylglycerols (DAGs) and monoacylglycerols (MAGs), while the complete hydrolysis involves to the formation of free fatty acids (FFAs) [29]. A detailed overview of such hydrolysis products in ancient pottery is elucidated in the next paragraph.

## 2. Lipids and Archaeological Biomarkers

In the last decades, the analysis of organic residues coming from foodstuffs, balms and perfumes found in pottery has been mainly focused on the determination of lipids. Archaeological biomarkers are specific molecules, often detected at trace levels, providing useful information on the origin of organic residues and clues about the potential function of the ceramic container in which they have been found [30,31]. In this section, the distribution and composition of the main lipid constituents detected in pottery such as acylglycerols (TAGs, DAGs and MAGs) and FFAs, and minor lipid constituents such as sterols (STs), natural waxes and terpenoids, will be discussed. The chemical structures of the most common lipids reported in the literature are illustrated in Figure 1.

### 2.1. Triacylglycerols, Diacylglycerols and Monoacylglycerols in Pottery

TAGs have sometimes been found in organic residues from pottery, since they usually readily decompose by means of chemical and microbial processes [13,32]. In the case of fats derived from ruminants (cattle, goats, sheep) and non-ruminant (pigs) animals, a narrow distribution of TAGs from C_42_, and from traces of C_44_ to C_54_–C_56_ carbon atoms has been detected, respectively. In dairy products from ruminant animals, a large distribution of TAGs from C_40_ to C_54_ was identified, whereas in marine and freshwater fish, they were not found [3,33].

As aforementioned, TAG decomposition reactions lead to the formation of DAGs, MAGs and FFAs. A low abundance of DAGs containing C_32_, C_34_, C_36_ long-chain acyl carbon atoms, together with significant concentrations of C_16_ and C_18_ MAGs, as well as C_40_–C_48_ wax esters, have been detected in archaeological pottery [33]. MAGs and DAGs containing C_16:0_ and C_18:0_ acyl moieties are degradation products of TAGs possibly present in raw animal fats [34]. The presence of these specific DAGs and MAGs, together with high concentrations of C_18:0_ and C_16:0_ FFAs, is representative of the degradation of animal fats [19] due to the use and/or subsequent burial of pottery for many centuries, as observed by Evershed et al. [35] by performing in-laboratory decay studies on animal fats.

### 2.2. Free Fatty Acids as Archaeological Biomarkers

FFAs are the principal constituents of hydrolyzed fats and oils, the most encountered and investigated lipid types associated with archaeological pottery [36]. Despite many FFAs being identified in archaeological ceramic sherds [37,38], only some of them were detected in significant amounts, especially if the ceramic containers have been treated at high temperatures for cooking purposes or they have been subjected to burial, being exposed to chemical reactions (oxidation, hydrolysis, condensation) [1,39]. FFAs consist, in most cases, of an unbranched hydrocarbon chain, mainly containing an even number of carbon atoms, commonly from 12 to 24, and a terminal carboxyl group. FFAs can differ from each other, not only in the carbon chain length, but also in the number of double bonds along the carbon chain. In such a respect, FFAs can be classified as saturated FAs (SFAs), monounsaturated FAs (MUFAs) and polyunsaturated FAs (PUFAs) [40].

The distribution of SFAs is strongly related to the nature of the organic residue detected in ceramics [30]. The most abundant medium chain SFAs found in pottery samples contain an even carbon number, such as palmitic (C_16:0_, hexadecanoic acid) and stearic (C_18:0_, octadecanoic acid) acids. These compounds are ubiquitous, since they can be identified both in animal and vegetable products [41]. On the other hand, lauric (C_12:0_, dodecanoic acid), arachidic (C_20:0_, eicosanoic acid) and behenic (C_22:0_, docosanoic acid) acids can be detected in significant amounts in coconut, palm and peanut oils, while myristic acid (C_14:0_, tetradecanoic acid) can be found in plant seed oils and dairy products [42]. Short chain FAs containing an even carbon number, namely butyric (C_4:0_, butanoic acid), caproic (C_6:0_, hexanoic acid), caprylic (C_8:0_, octanoic acid) and capric (C_10:0_, decanoic acid) acids, were identified in pottery in which ruminant milk fats, palm or coconut oil were contained [41]. SFAs with an odd carbon number such as pentadecylic (C_15:0_, pentadecanoic acid), margaric (C_17:0_, heptadecanoic acid) and nonadecylic (C_19:0_, nonadecanoic acid) acids were also revealed in ceramics [40,41]. Their origin is mainly linked to bacterial, milk and ruminant fats [43]. Short and medium chain SFAs with an odd carbon number, namely valeric (C_5:0_, pentanoic acid), enanthic (C_7:0_, heptanoic acid), pelargonic (C_9:0_, nonanoic acid), undecylic (C_11:0_, undecanoic acid) and tridecylic (C_13:0_, tridecanoic acid) acids, were detected in archaeological ceramics used as containers for the flowering plants valerian, rancid oils, pelargonium and other vegetable oils and dairy products, respectively [41]. In C_6_–C_24_ saturated fatty acids, as well as unsaturated FFAs, such as oleic (C_18:1ω9_, *cis*-9-octadecenoic acid) and linoleic acid (C_18:2ω6_, *cis*-9, *cis*-12-octadecadienoic acid) acids, a variable composition of C_14_–C_20_ alcohols, C_16_ and C_18_ MAGs, and C_23_–C_29_
*n*-alkanes was detected, together with small organic acids and monosaccharides deriving from glucose and glycerol, in pottery jars, vessels and amphorae possibly employed to store, contain and transport, at the same time or in different moments, vegetable oils or animal products with fermented alcoholic beverages (grape juice, wine) or sauces (Roman sapuum, mulsum or defrutum) [44,45,46]. Although tartaric and syringic acids were traditionally considered as wine biomarkers, the identification of wine in archaeological pottery remains controversial, since the aforementioned compounds can come from different sources [45,47,48,49]. Glutaric, fumaric, lactic malic, succinic and malonic acids, together with proper archaeological and historical support, could provide a more reliable interpretation of the data [46].

The Isotopic analysis of the δ^13^C values of the main C_16:0_ and C_18:0_ FFAs [50], their difference (Δ^13^C = δ^13^ C_18:0_– − δ^13^ C_16:0_) [3], as well as the proportions of selected SFAs [42], are particularly useful to obtain information about the origin of lipids determined in pottery [3,14,51,52,53,54,55,56,57,58]. Regert suggested that if the lipidic residue derives from non-ruminant animals, C_16:0_ and C_18:0_ FFAs are isotopically enriched in ^13^C with respect to those found in ruminants and Δ^13^C > −1‰, whereas goat adipose fats are featured by −3‰ < Δ^13^ C < −1‰ [3]. Whenever the source of the organic residue is a dairy product of ruminant animals, C_18:0_ acid is depleted in ^13^C with respect to adipose animal fats, and Δ^13^ C < 3.3‰. In the case of marine organisms, C_16:0_ and C_18:0_ FFAs are isotopically enriched in ^13^C with respect to those of terrestrial animals, although their values are not so different than domestic pig adipose fats. Freshwater fish resources are isotopically depleted in ^13^ C for both C_16:0_ and C_18:0_ acids with respect to marine fats [59]. Copley et al. observed organic residue displaying Δ^13^C > −1‰ which derive from ruminant fats, since the C_18:0_ free fatty acid is depleted in ruminant tissues because of bacterial processing in the rumen [60]. Shoda and colleagues confirmed the validity of the C_16:0_ and C_18:0_ FFA’s carbon isotopic variation criteria for the identification of lipids from ruminant animals. The same author also analyzed organic residues containing free fatty acids enriched in ^13^C, similar to those determined by performing analogous measurements on modern marine fish and salmonids [61]. This origin was further confirmed by the presence of a high (>80%) relative amount of the 3S,7R,11R,15-phytanic acid (SRR), typical of aquatic organisms. In the case of organic residues found in pottery vessels employed for cooking, containing and/or mixing different type of food such as lipids derived from acorns and chestnuts, freshwater fish, wild boar, wild ruminants and salmonids, the author [61] applied a concentration-dependent mixing model [62] taking into account δ^13^C_16:0_ and δ^13^C_18:0_ values together with the %SRR [63]. Dunne et al., investigating lipid residues found in ceramic bottles employed for childhood nutrition and determined Δ^13^C values between −3.4‰ and −3.7‰, which is attributable to the use of dairy ruminant products. Only in the case of one sample were the Δ^13^C values in the range between the dairy and non-ruminant fats, suggesting a possible mixing in the pottery vessel of pig or probably human milk with dairy products [64]. In another paper, Dunne and co-workers confirmed that C_16:0_ and C_18:0_ FFAs δ^13^C values are particularly useful to gain information on biosynthetic and dietary origin of fats detected in pottery from different ancient periods. The author observed lipidic residues deriving from non-ruminant animals featured by δ^13^C_16:0_ and δ^13^C_18:0_ values in the ranges from −11.0‰ to −28.1‰ and −11.0‰ to −26.9‰, respectively, and 0.1‰ < Δ^13^ C < 7.4‰. A ruminant adipose origin was attributed to samples displaying δ^13^C_16:0_ and δ^13^C_18:0_ values from −14.0‰ to −28.8‰ and Δ^13^ C ≤ −0.9‰. Dairy fats were identified with −3.0‰ < Δ^13^ C < 5.7‰, mixed ruminant and non-ruminant fats were featured by Δ^13^ C values in the range from −0.1‰ to −0.5‰ and mixed dairy and adipose fat with δ^13^C_16:0_ and δ^13^C_18:0_ values from −22.4‰ to −27.3‰ and −25.5‰ to −30.0‰, respectively, and Δ^13^ C = −3.2‰ [65]. Whelthon and co-workers asserted a comparison between the δ^13^C of archaeological C_16:0_ and C_18:0_ FFAs and modern reference animal fats can be considered reliable for animal fats only. Conversely, free fatty acids coming from plant product processing can cause a depletion in ^13^C and influence the δ^13^C C_16:0_ and C_18:0_ values. The author also stated that FFAs featured by a ^13^C enrichment or depletion for both the C_16:0_ and C_18:0_ could derive from the marine or freshwater commodity processing, respectively [18].

The investigation of the C_16:0_ and C_18:0_ FFA ratio is correlated to the study of the effect of environmental conditions on the FA degradation. Accordingly, some factors such as temperature, humidity and oxygen presence, are useful for the determination of degradation processes and their influence on the FFA concentration [42]. Notarstefano et al. suggested that whenever the C_18:0_ content is much higher than the C_16:0_ one (0.2 < C_16:0_/C_18:0_ < 0.6), the organic residues detected in pottery may come from herbivore animals [66]. If the amount of C_18:0_ is slightly higher than C_16:0_ (0.9 < C_16:0_/C_18:0_ < 1.3), the residues may have an animal origin. On the contrary, a C_18:0_ level slightly lower than the C_16:0_ amount (1.2 < C_16:0_/C_18:0_ < 2.0), may indicate a vegetable residue that, in the presence of long chain alcohols, may also suggest the presence of waxes. Gregg and Slater indicated that when the C_16:0_/C_18:0_ value is between 1.0 and 2.0, the residues could contain decomposed animal fats, while if its value is higher than 3.0, they may come from vegetable oils [53]. Kimpe and co-workers stated that when the ratio of C_16:0_/C_18:0_ is 1.0 ± 0.1, it means that the pottery vessels were employed for cooking purposes [54]. However, this conclusion was based on the analysis of only two vessel samples, and consequently should be considered with caution. The strategy of using the C_16:0_/C_18:0_ ratio as an archeological biomarker to evaluate the residue origin in pottery has not been completely accepted by some authors [3,67], since significant amounts of C_16:0_ and C_18:0_ FAs could also be the result of the conversion of unsaturated FAs to SFAs. Sikorski considered the interpretation of data based only on the C_16:0_/C_18:0_ ratio doubtful, since they can occur in high concentration either in plant or in animal sources [67]. Whelton et al. suggested a cautious interpretation of the calculation of the C_16:0_/C_18:0_ ratio coming from archaeological fats with respect to modern ones, due to the different solubilities and volatilities of FFAs, which could interfere with the correct ratio value estimation [18]. Other authors suggested that further FFAs or different ratios could provide a more reliable and accurate identification strategy [3,55,57,68,69]. Regert observed that, if the concentration of C_16:0_ is lower than C_18:0_, and a small concentration of C_15:0_ and C_17:0_ FFAs is also determined, together with oleic acid and its isomers, the organic residue may come from ruminant fats, including dairy products [3]. On the contrary, if the C_16:0_ content is higher than C_18:0_ and long chain FFAs containing three double bonds are detected, the organic residue could be related to fish fats. Olsson and Isaksson [57,70] proposed a C_18:0_/C_16:0_ inverse ratio combined with long-chain FA presence. This condition could suggest the nonvegetable nature of organic residues. Based on this concept, a C_18:0_/C_16:0_ ratio lower than 0.48 could be indicative of fish residues [57], whereas a value of C_18:0_/C_16:0_ higher than 0.48 could reveal the occurrence of decomposed fat in terrestrial animals [71]. In addition, Marchbanks and Malainey [55,69] proposed detailed ratios and indices among FAs for determining the nature of organic residues in archaeological pottery. For example, the percentage ratio (C_12:0_ + C_14:0_)/(C_12:0_ + C_14:0_ + C_18:2ω6_ + C_18:3ω3_) is less than 18% for vegetable oils, between 22% and 39% for fish fats, and higher than 47% for terrestrial animal fats. Analogously, Malainey [69] considered the FFA ratio (C_15:0_ + C_17:0_)/(C_12:0_ + C_14:0_ + C_16:0_ + C_18:0_) to discriminate against monogastric and ruminant animal fats residues. Eerkens and co-workers [68] stated that when this ratio exceeds 0.04, the organic residue may derive from ruminant fats. Last but not least, a C_17:0_ branched/C_18:0_ ratio was suggested by some authors [42,72,73] as archaeological biomarkers in order to identify organic residues coming from monogastric animal and ruminant fats.

Naturally occurring unsaturated FAs are featured by one to six double bonds along the carbon chain, in most cases with a *cis* configuration [40]. MUFAs are mainly distributed in plant oils such as olive, sesame and sunflower oils, or in avocados, peanuts, almonds, pecans, walnuts and cashews, while PUFAs are found in plant-based foods, oils and fish (trout, salmon, herring) [30,74]. The most common MUFAs detected in archaeological ceramic samples are palmitoleic acid (C_16:1ω7_, *cis*-9-hexadecenoic acid) and oleic acid. In the case of adipose fats derived from ruminant animals, a mixture of isomers of oleic acid with a double bond at the C_9_, C_11_, C_13_, C_14_, C_15_ and C_16_ positions has been also detected. In porcine fats and dairy products from ruminant animals only a single isomer of the oleic acid has been found as an unsaturated FA [3]. In addition, *cis*-vaccenic acid (C_18:1ω7_, *cis*-11-octadecenoic acid) was identified in milk and ruminant fat residues and erucic acid (C_22:1ω9_, *cis*-13-docosenoic acid) in rapeseed and mustard oils. Elaidic acid (C_18:1ω9_, *trans*-9-octadecenoic acid) was also detected in archaeological pottery used as containers for hydrogenated fats. The most abundant PUFA identified in archaeological ceramics is the already mentioned linoleic acid, which can be found in vegetable oils residues [41]. With respect to fish oils, FFA fingerprints are quite complex since SFAs (C_14:0_, C_16:0_ and C_18:0_), MUFAs (C_16:1ω7_, C_18:1ω9_) and PUFAs containing 18, 20 and 22 carbon atoms with a high degree of unsaturation (and up to six double bonds) have been reported [75]. Considering the high degree of unsaturation of PUFAs in fish oils, the probability that these molecules may degrade over time (by both chemical and biological degradation) is significant [13,76,77]. This means that a loss of unsaturated FAs over SFA compounds can be expected, hampering their possible detection in pottery [3,18]. In marine and freshwater fish, isomers of ω-(o-alkylphenyl) alkanoic acids (APAAs) (see Figure 2) with 16, 18 and 20 carbon atoms and positional isomers could be produced by degradation of tri-unsaturated FAs [3]. This was explained in terms of a multi-step alteration process, starting with the alkali isomerization of the acids, very likely promoted by pottery clays, followed by a 1,5-hydrogen shift with the formation of a conjugated triene system. Then, a *cis*/*trans* isomerization and an intramolecular Diels–Alder mechanism with aromatization may occur and produce conjugated cyclic products. Alternatively, a 1,7-hydrogen shift, an intramolecular Diels–Alder reaction and a final step of aromatization could lead to the formation of the mentioned products [22]. Vicinal dihydroxy acids are other oxidation products of unsaturated FAs. The position of the hydroxyl groups along the carbon chain indicates the original double bond position in the FA precursor. Such molecules, containing from 16 to 22 carbon atoms, have been detected in ancient pottery [78]. Furthermore, high concentration levels of isoprenoid fatty acids (IFAs), such as pristanic acid and phytanic acid, depicted in Figure 2, and low levels of 4,8,12-trimethyltridecanoic acid (4,8,12-TMTD), can be found in marine animals. On the other hand, they are not present in terrestrial animals [3,37]. The branched structures of IFA compounds are particularly resistant to degradation; therefore, they are listed as archaeological biomarkers for the identification of fish oils contained in pottery vessels [79,80]. In the case of adipose ruminant and non-ruminant animal fats, oxidation reactions could cause the formation of unsaturated short-chain dicarboxylic, hydroxy- and dihydroxy carboxylic acids, both in the free and esterified form [19]. The detection of (α,ω)-dicarboxylic acids ranging from C_5_–C_7_ to C_12_–C_13_ has been reported in the literature [38]. The azelaic acid (nonanedioic acid, Figure 2) represents one of the most detected (α,ω)-dicarboxylic acids, indicating that FA precursors were featured by a double-bond at the C_9_-position (i.e., oleic acid). Alkaline hydrolysis of FFAs could also produce ω-hydroxy even-numbered saturated carboxylic acids with 8–12 carbon atoms. Heating animal fats could induce the condensation of FFAs and the formation of odd-numbered monounsaturated ketones ranging from C_29_ (nonacosan-15-one, Figure 2) to C_33_ (tritriacontan-16-one)-C_35_ (pentatriacontan-18-one) [3].

### 2.3. Minor Lipid Constituents in Archaeological Samples

Phospholipids (PLs) are structural components of biological cell membranes [8]. They consist of a phosphoric acid unit, often linked to a nitrogen-containing molecule and two FAs [81,82]. PLS are the constituents of the carcass fat of wild ruminants with higher concentrations with respect to the acylglycerols [83]. In soil, where pottery sherds are usually found after prolonged burial, PL’s occurrence is transitory due to degradation processes that make them difficult to detect after many centuries, so that the only extractable compounds belonging to this class are hypothesized to derive from living biomass [83].

STs are precursors of some hormones and structural components of cell walls [30]. Their low concentrations in pottery, as well as the possible cross-contamination due to the handling of pottery during the excavation or post excavation phases, make the STs identification particularly questionable [18]. STs can be mainly grouped into phytosterols and zoosterols based on vegetable and animal origin, respectively [8]. Phytosterols, such as sitosterol, stigmasterol and campesterol, and their oxidation products such as sitostanone (Figure 2), sitostanol and campestanol, can be identified as organic residues coming from vegetable oils, cereal grains and nuts. For this reason, they are considered plant ST biomarkers in archaeological potsherds [42]. They were sometimes identified in visible carbonized residues on the inner surface of pottery vases or in the botanical remains of carbonized seeds [42,66]. The most abundant zoosterol is undoubtedly cholesterol, a component of the biological membranes of mammal cells and a precursor of estrogen and steroid hormones (progestogens, glucocorticoids, androgens, mineralocorticoids) [84]. Cholesterol either has an exogenous source, coming from animal food, and/or an endogenous origin, because it is produced in the animal liver [84]. Whelton et al. stated that its detection should be viewed with caution due to a possible contamination from human skin lipids [18]. Only if cholesterol is determined together with its hydroxy-, oxo-, epoxy, ketone oxidized derivatives, such as cholestanol and cholestanone (Figure 2), produced by the heating of animal fats in pottery vessels or the natural decay, can it be considered as an archaeological biomarker [33,85]. Saturated odd-numbered mid-chain ketones from C_29_ to C_35_, such as nonacosan-15-one (Figure 2), triacontan-14-one, triacontan-15-one, hentriacontan-16-one, dotriacontan-15-one, dotriacontan-16-one, tritriacontan-16-one, tetratriacontan-17one and pentatriacontan-18-one, as well as C_33_ and C_35_ monounsaturated ketones, were also detected in pottery containers where animal fats were processed [3]. Ergosterol is a specific mycosterol, component of fungal cell membranes with the function of cholesterol in animals and precursor of ergocalciferol (vitamin D2). In lipid residues found in prehistoric pottery, it was considered a potential biomarker for alcohol fermentation in beer, bread or wine [42], but Isaksson et al. highlighted that ergosterol detection in pottery could also derive from modern contamination [71].

Other lipids identified in organic residues extracted from pottery could include natural waxes biosynthesized by insects, such as honey and beeswax, or by plants, that form hydrophobic coatings on their outer surface, as in the case of leaf or epicuticular waxes, and protective layers on the skin, hair and feathers of animals [3,30,74,82,86]. Wax composition is heterogeneous and varies with plant or animal type [87]. Waxes from leafy plants are characterized by long chains archaeological biomarkers, such as odd-numbered alkanes (C_25_–C_33_), even-numbered alcohols (C_20_–C_34_) and aldehydes (C_24_–C_28_), as well as C_39_–C_52_ esters [88,89]. Some organic compounds are considered to be specific biomarkers of plant oils as a source of organic residues, as in the case of *Brassicaceae* seed oil, widely employed in ancient times, whose chemical fingerprints are gondoic acid (C_20:1ω9_, *cis*-11-eicosenoic acid) and the aforementioned erucic acid. Oxidation processes occurring on these compounds could produce vicinal dihydroxy acids and (α,ω)-dicarboxylic acids. Vicinal dihydroxy acids such as 11,12-dihydroxy arachidic (11,12-dihydroxyeicosanoic acid) and 13,14-dihydroxybehenic (13,14-dihydroxydocosanoic acid) acids, as well as the (α,ω)-undecanedioic and (α,ω)-tridecanedioic acids, are chemical fingerprints of *Brassicaceae* seed oils [29]. *n*-Nonacosane (Figure 2) and its oxygenated derivatives nonacosan-15-one and nonacosan-15-ol are indicators of the processing of cabbage, turnip, kale and broccoli vegetables [30,89]. In waxes, the formation of soluble salts and volatilization reactions caused by heating of the ceramic vessel, could produce loss of FAs and of *n*-alkanes, respectively [82]. Beeswax is featured by long-chain odd-numbered alkanes (C_23_–C_33_), even-numbered FAs and even-numbered C_40_–C_54_ wax esters. Esters are more resistant to hydrolysis than TAGs so that wax can be considered less vulnerable to degradation processes and more likely detectable in archaeological samples with respect to TAGs [87]. Partial degradation products could occur over time on beeswax esters and produce long-chain even-numbered alcohols (C_24_–C_34_), FAs and n-alkanes.

Lipids detected in pottery could also originate from natural products not employed as foodstuffs, such as those coming from resins, tars, pitches and bitumen and wax from beeswax used for non-dietary purposes. These products were stored in vessels, used as sealants, for decoration, as adhesives for repair aims, as illuminants, ointments, cosmetics, balms and medicines. The main lipids detected in these matrices are terpenoids, whose structures are featured by isoprene (C_5_H_8_, 2-methylbutadiene) units that can be classified in monoterpenoids (C_10_), sesquiterpenoids (C_15_), diterpenoids (C_20_) and triterpenoids (C_30_) with 2, 3, 4 and 6 isoprene units, respectively. Terpenoids generally show a good preservation [90], even if their degradation products were detected in archaeological pottery samples. The loss of low molecular weight terpenoids, such as monoterpenoids and sesquiterpenoids, was observed due to their volatility [82]. Diterpenoids such as dehydroabietic and didehydroabietic acids, as well as their oxidation products, namely 7-oxo-dehydroabietic (Figure 2), 7-oxo-abietic and 15-hydroxydehydroabietic acids, are considered specific archaeological biomarkers of *Pinaceae* family resins [66,91,92]. Some of these compounds can provide information on the resin heating conditions, as for methyl dehydroabietate, which indicates that pine resin was heated at high temperatures and in the presence of wood [91]. Such molecules are often detected together with wine-related compounds such as tartaric acid, since resins were employed as sealing or waterproofing agents in pottery vessels, and for wines aromatization. In addition, pine resins were used in firing containers to improve the mechanical and thermal resistance of ceramic to heating [66]. Triterpenoids such as oleanonic and oleanolic acids, together with other compounds, are chemical fingerprints of storax resin [93]. Betulin, lupeol and their derivatives, such as lupenone, betulone (Figure 2) and betulinic acid, are archaeological biomarkers of birch bark [94]. In the last decade, a pentacyclic triterpene methyl ether named miliacin, commonly found in millet grains, was also identified in prehistoric pottery vessels [38,95].

## 3. Artificial Ageing Studies

In-laboratory alteration processes such as thermal decomposition, oxidation and hydrolysis of lipids have been investigated in order to simulate the natural degradation occurring in archaeological contexts. Such ageing studies can provide key elements to better interpret the origin of animal fats and plant oils that are partially or totally altered over time. The elucidation of the chemical and biochemical mechanisms responsible for the alteration of pristine molecules also allows for the unveiling of the life history of an organic residue. For example, long-chain ketones are formed via free radical-induced dehydration and decarboxylation mechanisms, which involve intensive heating of the carboxylic FAs up to over 300 °C [25]. As a result of such cooking activities, the degradation of unsaturated FAs over 270 °C with formation of APAAs was also reported. This means that their detection may provide clues on the original constituents fired in pottery.

FAs containing at least one double bond along the carbon chain are particularly sensitive to oxidation reactions. The oxidation process involves the inclusion of an oxygen atom in the carbon chain, the scission of the double bond and formation of lower molecular weight species [29]. As reported by Rottlander and Schlichtherle, the oxidation rate of FAs depends on the degree of unsaturation. This means that MUFAs are oxidated much more slowly than PUFAs [77]. (α,ω)-Dicarboxylic and ω-hydroxycarboxylic acids are often the main oxidation products, as reported by Colombini et al. [29]. The authors performed ageing experiments by heating gondoic and erucic acid standards at 120 °C for three weeks [29]. Gas chromatography–mass spectrometry (GC–MS) analyses revealed that oxidation products were strongly influenced by the double bond position along the carbon chain. Accordingly, (α,ω)-undecanedioic and (α,ω)-tridecanedioic acids were the most abundant oxidation compounds and their formation via radical oxidation mechanism was consistent with the gondoic and erucic structures, respectively. The GC–MS chromatograms, reported in Figure 3, also highlighted the presence of minor constituents with short- and medium-chains, indicating further reaction mechanisms such as migration of the radical adjacent to the carboxylic group [29]. Alternatively, Bondetti and co-workers focused their attention on the study of APAA species as degradation products of MUFAs and PUFAs [96]. Their formation was the result of double-bond rearrangements during protracted heating of lipids present in animal and plant tissues. The carbon chain length of APAAs allowed the organic residues coming from aquatic or terrestrial source to be distinguished. In fact, the presence of APAA with 20 and 22 carbon atoms could be directly related to the cooking of aquatic organisms such as freshwater and marine animals [97], since they derive from their long-chain FA precursors, eicosapentaenoic (C_20:5ω3_, eicosapentaenoic acid or EPA) and docosahexaenoic acid (C_22:6ω3_, docosahexaenoic acid or DHA), which are quite common in aquatic sources. In contrast, APAA-C_16_ and APAA-C_18_ species can be considered as archaeological biomarkers for revealing the terrestrial nature of the organic residue, belonging to both the animal and the plant kingdoms. In addition, the simulation of the degradation reactions allowed the authors to establish a further threshold value of 0.06 for the ratio APAA-C_20_/APAA-C_18_, in order to discriminate aquatic sources from terrestrial products. Moreover, Hammann et al. recently carried out artificial ageing of cholesterol in ancient clay potteries [84]. The authors demonstrated that cholesterol undergoes complete degradation at 100°C in the presence of a high content of FAs, consistent with those observed in animal fats (100:1 *w*/*w*, FA to cholesterol). However, the pro-oxidative behavior of FAs had a minimal effect on the cholesterol degradation when low concentration levels were registered (1:4 *w*/*w*, FA to cholesterol). In-laboratory heating experiments suggested that the clay contributed to the cholesterol degradation. In fact, different degradation products were observed during the experiments involving cholesterol heating with the clay material only. Since both FAs and the clay surface may contribute independently to degradation reactions, the absence of cholesterol in cooking lipid residues in ancient pottery is not surprising. Within such a context, the detection of cholesterol in clay pottery should be examined with caution [84].

The alteration of lipid substances over time is also partly mediated by bacterial action. In fact, it is well known that bacteria adapt readily where essential nutrients, including lipids, are available. Consequently, the effects of bacterial activity on lipids in archaeological organic residues have been discussed in the literature. Dudd et al. designed laboratory experiments in order to simulate the decay of absorbed lipid matter in ceramic vessels under oxic conditions [98]. In detail, two of the most popular foods in antiquity, milk and olive oil, were absorbed on sherds and incubated at 30 °C in a flask with mushroom compost (mushroom humix manure). The decay of lipid compounds was monitored at different times intervals. In-laboratory ageing experiments indicated that the degraded lipid profile of milk was indistinguishable from that of adipose fat. Extreme caution is therefore required in the assignation of such lipid matter. In addition, the bacterial-induced decay of lipids shows typical biomarkers such as branched-chain and odd carbon number FAs. However, such compounds naturally occur in milk fat due to the presence of bacteria in the rumen [99], thus the bacterial action on lipid degradation cannot be established with sufficient certainty [98]. On the other hand, the simpler lipid profile of olive oil allowed the authors to assess the contribution of bacteria to the decay of the original lipids, presumably resulting from the combination of microbiological and abiological hydrolysis [98]. In fact, the detection of branched chain FAs in olive oil indicated that bacterial organisms were actively responsible for the decay of the olive oil in potsherds, although to a lesser extent than other degradation mechanisms.

## 4. Sampling and Extraction Protocols of Lipids from Ancient Pottery

The capability of elucidating the lipid composition of materials used by the ancient societies can ensure that their practices remain part of our cultural heritage. Consequently, the chemical analysis of preserved lipid matter in archaeological contexts is not a simple and routine procedure, but it requires careful planning of the entire analytical workflow, from the sampling strategy to the interpretation of the analytical data. For a successful research study, a project design sampling strategy is the first and crucial aspect that must be emphasized. In fact, a significant criticism of the sampling methods has already been highlighted in the literature [8,18]. It seems quite clear that a single vessel or limited number of vessels cannot provide meaningful data, except for minor circumstances such as an archaeological sample coming from a “special depositional context” or showing “special typological characteristic” [18]. This means that a robust sampling strategy involving a large number of sherds (20–30) is necessary in order to statistically represent the time period, excavation site, burial conditions, object shape and pottery typologies. A large number of samples also enables the examination of a range of potential variables that may affect the quality of analytical data [8,18]. Another fundamental aspect involves the strict collaboration between the analyst and the archaeologist, essential to ensure a coherent strategy to the whole process. For example, the archaeological area can provide preliminary relevant information regarding the context and relationships within and between sites, as well as the absolute and relative chronological information, pottery typologies, materials etc. [18]. If such information is not included in the project design sampling strategy, the entire research may be of questionable quality. Appropriate handling and storage protocols are the key requirements to avoid the presence of contaminants. In general, the first source of contamination is linked to the organic matter in the burial environment. Animal and plant materials, as well as microbial synthesis from bacteria and fungi, represent all potential interfering agents that should be critically addressed. In order to exclude any possible lipid contamination, it is advisable to perform a comparison of the analytical data between the pottery excavated from the burial site and the surrounding soil. The effectiveness of such an approach was demonstrated by Heron et al. [100]. Contaminant agents can also come from handling of the sherds, both during excavation and post-excavation [18]. For instance, the human contact introduces lipid contaminants including cholesterol and squalene that can be mistaken with animal organic residues preserved in the vessels. Squalene is known to degrade over time; therefore, its detection is usually assigned to modern handling. On the other side, as aforementioned, the presence of cholesterol should be interpreted with caution due to its capability of surviving in organic residues of pottery, except when exposed to a high firing temperature [100]. Contaminants can also arise from materials used for the storage of samples such as phthalate plasticizers. In this case, paper bags, rather than plastic bags, solve the problem of plasticizer contamination, although these substances are easily identifiable and do not affect the analysis of lipids on pottery [8]. As routine practices, the monitoring of a blank sample is crucial for the detection of lipid contaminants. This measure allows not only for the examination of the purity of the chemicals (solvents and reagents), but also to evaluate the entire analytical protocol including the chromatographic instrumentation. Based on this assumption, the pottery analyses should be performed when a blank, without any lipid contaminants, is obtained. Of course, this approach can only be used to establish contaminants introduced in the laboratory, not those introduced through handling and storage protocols.

Organic residues in archaeological pottery represent very complex matrices, therefore it is necessary to optimize an adequate sample treatment. Generally, lipids are analyzed by using chromatographic techniques coupled to an MS detector [101] in order to reveal the correct identity of the lipids in a univocal manner. The extraction step is mandatory in order to purify and isolate the lipid compounds. As a rule, there is not a univocal protocol of sample preparation that is suitable and applicable for all archaeological matrices and compatible with all the analytical techniques. For this reason, numerous sample treatments have been described in the literature over the years for lipid analysis in pottery.

In term of lipid extraction, the literature data confirm that solvent-based extractions are the most common. Recently, Whelton et al. reported that chloroform:methanol (2:1 by volume) and dichloromethane:methanol (2:1 by volume) solvent mixtures have been used for the extraction of lipids from ceramics [18]. In these solid–liquid extraction techniques, widely utilized in foodomics and lipidomics research, lipid compounds are simultaneously extracted and isolated into a liquid layer. For example, Harper et al. utilized 5 mL of a 2:1 *v*/*v* dichloromethane:methanol solution to extract lipids from pottery [102]. Evershed et al. extracted lipids from powdered sherds (2 g) by using a chloroform:methanol (10 mL, 2:1 by volume) solvent mixture [103]. Both protocols demonstrated to be suitable in the extraction of FFAs, MAGs, DAGs and TAGs from archaeological samples. An alternative extraction strategy has recently been optimized by Tanasi et al. [104]: a solution of chloroform:methanol:water (1:1:0.9) was added to suspend the powder and extract “free” lipids. Compared to the waterless extraction procedures described above, such strategy requires that lipid compounds and interfering substances are partitioned into two layers: the upper methanolic phase containing virtually all of the non-lipid substances and lower chloroformic phase consisting of lipids [105]. From a methodological point of view, such a protocol requires the use of a centrifuge to allow the clear separation of the biphasic system. In order to improve the extraction of lipids, some authors also introduced sonication, as described by Regert et al. [3].

Alternative solvent-based extraction protocols have also been reported in the literature, especially whenever the recovery of polar lipids such as dicarboxylic acids and hydroxyl acids was incomplete, by using the well-established chloroform:methanol extraction. In fact, lipid molecules containing one or more polar functional groups, such as carboxyl (–COOH) and hydroxyl (–OH) portions, form strong intermolecular interactions (i.e., hydrogen bonds, dipole–dipole, ion–dipole and electrostatic interaction) with the polar surface of the ceramic matter. This means that their removal/extraction from ceramic is favored by more aggressive alkaline or acid reagents. The effectiveness of the alkaline treatment was demonstrated by Regert et al. [34]. In detail, a portion (1 g) of sherd, already subjected to a solvent-based extraction, was re-extracted with 10 mL of NaOH methanolic solution (0.5 M) at 70 °C for 90 min. Alkaline treatment confirmed the presence of (α,ω)-dicarboxylic acids (from C_7_ to C_12_) and α-hydroxy carboxylic acids (from C_8_ to C_12_) in the total lipid composition of organic residues that were not extractable by using a chloroform/methanol solvent mixture. The capability to extract the most polar lipids was also evaluated by using acidified methanolic solution (H_2_SO_4_–MeOH, 2% by volume) [51]. An alternative extraction technique is represented by the microwave-assisted extraction (MAE). One of the main advantages of this strategy is the reduction in the extraction time due to the different mechanism, by which heat is transferred to the solution. In fact, the use of microwaves allows the rapid heating of the solution, keeping the temperature gradient to a minimum. This actually increases the heating speed of the extraction mixture [106]. Although the MAE technique has great potential for heating materials, the correct selection of the extraction solvents is the fundamental requirement for a successful application. As a general principle, a solvent utilized in the conventional extraction procedures is not suitable in the MAE process if it is not able to absorb microwave energy. The use of MAE in the archaeological field has recently been reported by Blanco-Zubiaguirre et al. [24]. The extraction of TAGs was performed by mixing 1 g of the archaeological ceramic sample with 900 µL of the chloroform:hexane solution (3:2 by volume). The power of microwaves was fixed to 600 W for 25 min (temp. 80 °C). The authors also performed the extraction and simultaneous saponification of FAs by using a KOH ethanolic solution (10% *w*/*v*). In this case, the power of the microwave was 200 W for 60 min (temp. 80 °C). Other authors have performed the microwave-based extraction of lipids from ancient pottery [53,107], but the MAE approach is not widespread in archaeometry yet.

Last but not least, an innovative extraction technique based on the use of supercritical fluids has been recently explored by Devièse et al. [108]. As reported by the same authors, supercritical fluid extraction (SFE) has not yet been widely explored in the field of archaeological science, but the results of this pilot study are very promising. First of all, the SFE technique does not require the destruction of the sherd sample, since the lipid extraction can be performed without any further grinding step by pestle or mortar. This aspect is crucial for the reduction in the potential source of contamination that can substantially affect the analytical results. In addition, the methodology involves the use of supercritical carbon dioxide (CO_2_), an excellent extraction solvent due to its relatively high density and consequently high solvation power. Further, it presents a low viscosity and high diffusion coefficient that allows fast extraction. From the environmental point of view, the supercritical CO_2_ greatly reduces the use of organic solvents. In fact, only small portions of solvents are combined with the supercritical fluid in order to change its polarity. Devièse et al. [108] optimized the extraction of lipids from pottery by the SFE method using water and ethanol as cosolvents, which are less toxic than extraction solvents conventionally used in solid–liquid extraction. Regarding the extraction performance, the SFE allowed the authors to detect, in laboratory-made ceramic samples, higher quantities of lipids than those obtained by using a conventional solvent-based extraction (chloroform:methanol, 2:1 by volume) protocol. In addition, small amounts of unsaturated FAs were surprisingly detected in the SFE extracts, while they remained unrevealed using the solvent-based method. The SFE procedure thus showed a high extraction yield of lipids from ceramic containers, a fundamental requisite to establishing the correct functions of ceramic vessels. In light of the extraction procedures so far described, the solvent-based extractions are certainly the most widely used in the archaeological field, due to their reproducibility and reliability. However, these methodologies involve numerous sample preparation steps, sometimes more than necessary, that affect not only the duration of the entire analytical process, but also the quality of the analytical data in terms of extraction yield and contamination. With regard to the alternative and innovative strategies, both MAE and SFE techniques are still not particularly widespread for lipid analyses in archaeological pottery. As a general rule, the most innovative experiments take a long time to be accepted and used instead of the conventional procedures. Nevertheless, the effort required to migrate from a traditional extraction method towards an innovative one is counterbalanced by the equal or higher extraction yield that can be obtained in a shorter time. In addition, these alternative methodologies include practices that are in accordance with the miniaturization of chemicals consumption, preferential usage of low-toxicity reagents and a reduction in waste production.

Focusing on the total FA composition, widely elucidated by using GC techniques, a derivatization step is mandatory to convert the lipids into more volatile and less polar compounds [109]. In fact, if, on the one hand, high-molecular weight species such as FAs linked to glycerol (i.e., TAG), cholesterol or long-chain aliphatic alcohols (i.e., waxes) are not amenable for GC separation, on the other hand molecules containing functional groups such as –COOH or –OH (i.e., carboxylic acid, dicarboxylic acid, sterols, etc.) can form a hydrogen bond between compounds. This leads to poor volatility, insufficient thermal stability and low detectability due to the strong interaction between polar components and the stationary phase of the GC column [110]. Thus, the derivatization is a very useful procedure to modify the chemical structure of a compound and improve its chromatographic properties. In the archaeological field, different derivatization strategies are reported in the literature. The most common one is the saponification reaction followed by methylation or silylation, which is a classical method for the preparation of fatty acid methyl esters (FAMEs) from acylglycerols [111]. For example, Harper et al. performed the methylation of lipid residues by using sodium hydroxide (KOH) in a methanol solution (5 M) as basic catalyst [102]. The methylation reaction was carried out in an ultrasonic bath for 30 min. Based on the mechanism of transesterification (ester exchange), the acylglycerols were completely methylated into FAMEs. However, FFAs are not normally esterified by using a methanol solution in the presence of a basic catalyst [105]. In order to overcome this drawback, Romanus et al. investigated the FAME composition of archaeological potsherds by using a dual-stage derivatization approach [112]. In detail, the method involved the use of boron trifluoride (BF_3_) in methanol (50% *w*/*v*) as an acid catalyst and KOH in methanol (1 M) as a basic catalyst in order to simultaneously methylate acylglycerols and FFAs. The addition of BF_3_ was particularly advantageous, although methoxy artefacts can be produced by the addition of methanol across the double bond of FAs, especially in the presence of a very high concentration level of BF_3_ (i.e., 50% *w*/*v*) [105]. For this reason, it is strongly suggested to use lower concentrations of BF_3_ in methanol, in order to avoid the formation of the by-products that can alter the real FAMEs composition. In such a respect, some authors performed the lipid derivatization using a BF_3_ methanolic solution 14% *w*/*v* at 70 °C for 1 h [6,79]. The silylation reaction involves the replacement of a labile hydrogen from acids or alcohols with a trimethylsilyl (TMS) group and the formation of derivative esters. Compared to the methylation approach, silylation allows the derivatization of multiple functional groups, such as carboxyl or hydroxyl, in one step. One of the most common silylation reagents used is N,O-bis-(trimethylsilyl)trifluoroacetamide (BSTFA) with the addition of 1% trimethylchlorosilane (TMCS) [29,34,91,103].

Many procedures have been described in the literature, in which lipids are extracted and methylated at the same time [4,51,113,114]. Such a strategy is the so-called “direct extraction-derivatization”, which reduces the number of clean up-steps and is therefore particularly advantageous in terms of time and costs per analysis [109]. On the other side, the risk that non-lipid contaminants may interfere with subsequent GC analyses should be considered as well [105]. Such an approach was performed by Correa-Ascencio and Evershed [51]. The authors used an acidified methanolic solution (H_2_SO_4_-MeOH, 2% by volume) for extracting and simultaneously methylating lipid residues in powdered sherd samples. The developed strategy involved the heating of the reaction mixture at 70 °C for 1 h. Compared to the conventional solvent extraction (CHCl_3_–MeOH, 2:1 *v*/*v*), the direct methanolic acid treatment enhanced the recovery of lipid residues, which is an ideal condition considering the low concentration levels of the lipid residues in pottery. In addition, the use of the acidified methanolic solution allowed the authors to reveal the presence of some polar lipid compounds such as dicarboxylic acids and long-chain ketones, which were unextracted using the CHCl_3_–MeOH method. Both lipid extraction protocols were also evaluated by Reber [115]. The analytical workflows of the applied procedures are illustrated in Figure 4.

The effectiveness of the direct extraction derivatization procedure was also demonstrated by Demirci et al. [114] and Papakosta et al. [116]. Besides the typical lipid distribution found in archaeological pottery (i.e., FAs, acylglycerols), the acidified methanol treatment allowed the extraction of APAAs, with carbon atoms ranging from 18 to 22, and isoprenoid FAs, including pristanic and phytanic acid. Some lipid-based archaeological studies and related analytical strategies, including extraction and derivatization approaches, are listed in Table 1.

## 5. Conclusions

The study of lipid residues in pottery is a key element to ensure that the practices of ancient societies remain part of our cultural heritage. The detailed composition of the lipid matter can unveil crucial hints about daily life, diet, food storage and processing, ritual and medical practices, etc. A large number of studies discussed in the first part of this review deal with specific archaeological lipid biomarkers. Within such a context, it appears that TAGs, DAGs, MAGs and FFAs represent the main classes of lipids detected in pottery. Nevertheless, the importance of other lipid classes such as (α,ω)-dicarboxylic acids, APAAs, beeswax, sterols, etc. has been highlighted.

As far as lipid alteration over time, mediated by thermal decomposition, oxidation and hydrolysis reactions is concerned, several authors have simulated the natural degradation of lipids occurring in archaeological contexts in the laboratory. Such artificial ageing studies are of significative importance since they allow the clarification of the degradation mechanisms responsible for the decay of lipid structures. From the discussion reported in the second part of the review, it can be concluded that careful planning of the entire analytical workflow is the pivotal step for a successful research study in the archaeological context. An adequate sampling strategy must be developed in order to ensure the quality of analytical data. First of all, a considerable number of samples guarantees the reliability of the study, especially from a statistical point of view, because it takes into account a range of potential variables affecting the quality of the data. In addition, special care should be given to the handling and storage protocols to avoid the presence of potential contaminant agents, both during excavation and post-excavation. In general, the main rules to be applied have been clarified in this review article.

Finally, several analytical approaches useful for the lipid characterization of ancient pottery have been discussed. From the literature data, organic solvent-based extractions resulted in well-established protocols. However, despite the fact that this type of approach guarantees reliability of the analytical data, no particular developments have been made in term of innovation or miniaturization for the reduction of chemicals. In the light of the studies reported so far, the extraction strategies need further optimization in order to be competitive with the well-established methodologies currently in use.

## Figures and Tables

**Figure 1 molecules-27-03451-f001:**
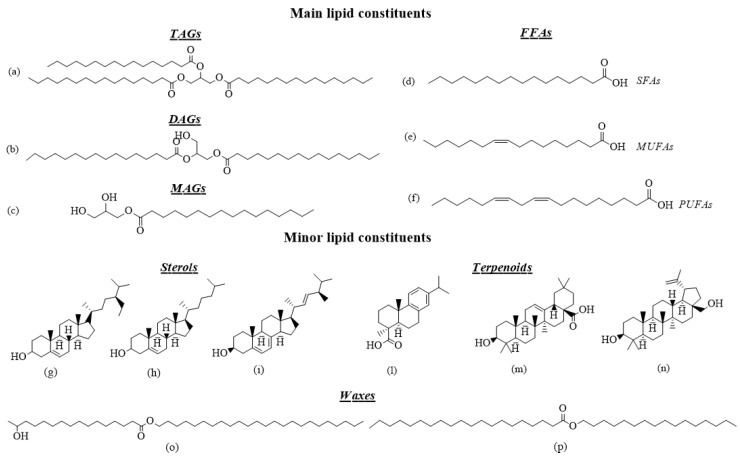
Examples of lipids detected in archaeological pottery samples: (**a**) tripalmitoylglycerol, (**b**) 1,3-dipalmitoyl-glycerol, (**c**) 1-palmitoyl-glycerol, (**d**) palmitic acid, (**e**) palmitoleic acid, (**f**) linoleic acid, (**g**) sitosterol, (**h**) cholesterol, (**i**) ergosterol, (**l**) dehydroabietic acid, (**m**) oleanolic acid, (**n**) betulin, (**o**) tetracosanyl 15-hydroxypalmitate, (**p**) hexadecyl eicosanoate.

**Figure 2 molecules-27-03451-f002:**
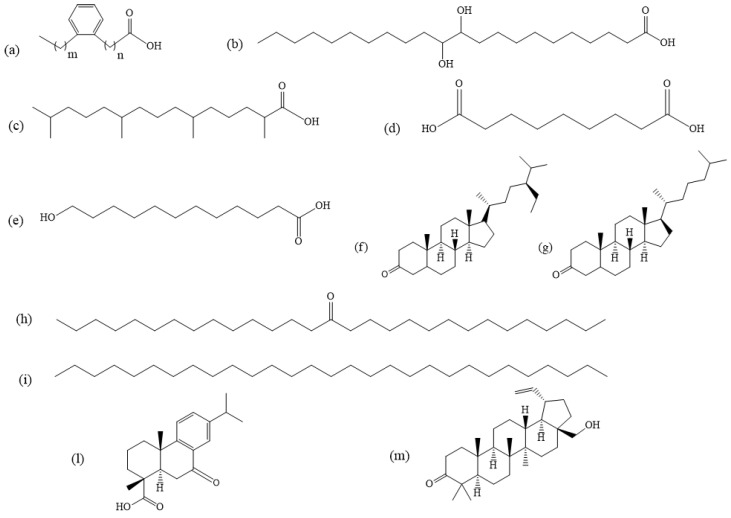
Examples of lipids degradation products detected in archaeological pottery samples: (**a**) ω-(ο-alkylphenyl)alkanoic acid, (**b**) 11,12-dihydroxydocosanoic acid, (**c**) pristanic acid, (**d**) azelaic acid, (**e**) ω-hydroxydodecanoic acid, (**f**) sitostanone, (**g**) cholestanone, (**h**) nonacosan-15-one, (**i**) *n*-nonacosane, (**l**) 7-oxodehydroabietic acid, (**m**) betulone.

**Figure 3 molecules-27-03451-f003:**
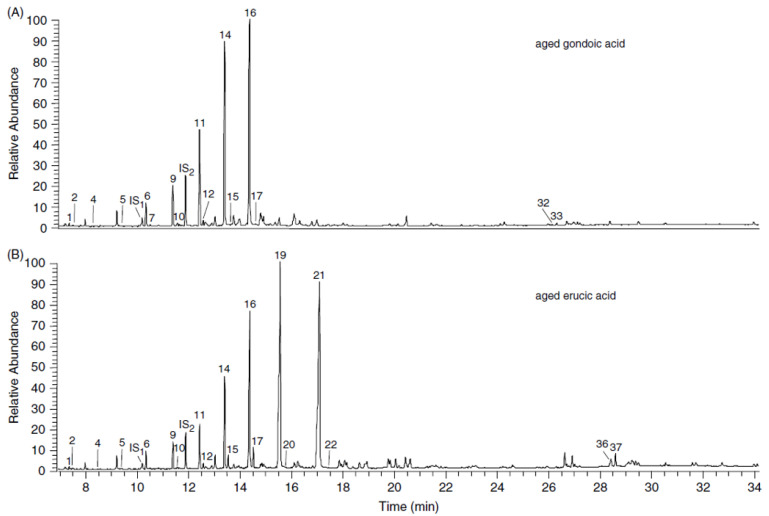
GC–MS chromatogram of aged gondoic (**A**) and erucic (**B**) acid standards. Reprinted with the permission from Ref. [91]. Copyright 2005 John Wiley and Sons, Ltd. Peak assignment is described as follows: (1) nonanoic acid, (2) (α,ω)-butanedioic acid, (3) decanoic acid, (4) (α,ω)-pentanedioic acid, (5) (α,ω)-hexanedioic acid, (6) (α,ω)-heptanedioic acid, (7) ω-hydroxyoctanoic acid, (8) dodecanoic acid, (9) (α,ω)-octanedioic acid, (10) ω-hydroxynonanoic acid, (11) (α,ω)-nonanedioic acid, (12) ω-hydroxydecanoic acid, (13) tetradecanoic acid, (14) (α,ω)-decanedioic acid, (15) ω-hydroxyundecanoic acid, (16) (α,ω)-undecanedioic acid, (17) ω-hydroxydodecanoic acid, (18) hexadecanoic acid, (19) (α,ω)-dodecanedioic acid, (20) ω-hydroxytridecanoic acid, (21) (α,ω)-tridecanedioic acid, (22) ω-hydroxytetradecanoic acid, (23) oleic acid, (24) octadecanoic acid, (25) (α,ω)-tetradecanedioic acid, (26) gondoic acid, (27) eicosanoic acid, (28) 9,10-dihydroxyoctadecanoic acid, (29) 9,10-dihydroxyoctadecanoic acid, (30) erucic acid, (31) docosanoic acid, (32) 11,12-dihydroxyeicosanoic acid, (33) 11,12-dihydroxyeicosanoic acid, (34) nervonic acid, (35) tetracosanoic acid, (36) 13,14-dihydroxydocosanoic acid and (37) 13,14-dihydroxydocosanoic acid. All compounds are intended as TMS derivatives.

**Figure 4 molecules-27-03451-f004:**
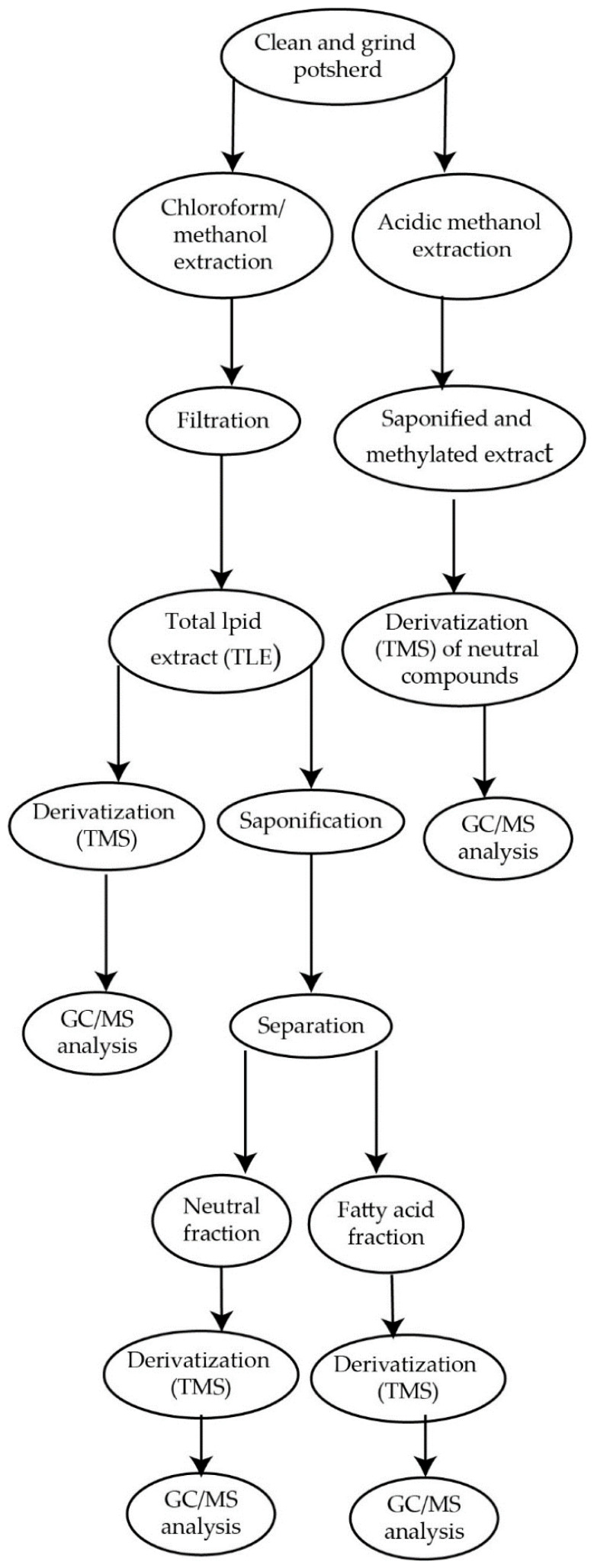
Analytical workflows of both conventional solvent extraction (chloroform/methanol) and direct extraction derivatization protocols. Reproduced with kind permission of MDPI [115].

**Table 1 molecules-27-03451-t001:** Examples of lipid-based archaeological studies and relative analytical approaches.

Pottery Samples and Archaeological Site	Lipid Biomarkers	Extraction	Derivatization	Analysis Method	Probable Origin	Ref.
n.35 from Zamostjen n.2 from Joton n.20 from Tianluoshan	APAAs	Solvent Extraction: MeOH/H_2_SO_4_ (4 h at 70 °C)	Direct extraction- derivatization	GC-MS GC-MSD	Aquatic	[96]
n.6 from Western Iberian Peninsula	ω-Hydroxy acids and cholesterol	Solvent Extraction: CH_2_Cl_2_/MeOH (2:1)	BSTFA + 1% TMCS *	GC-MS	Beeswax	[117]
n.14 from George Reeves, Mississippi Valley	Sterols, alkanols, alkanes and terpenoids	Solvent Extractions: CHCl_3_/MeOH (2:1) and MeOH/H_2_SO_4_	BSTFA + 1% TMCS (70 °C for 1 h) NaOH in methanol (75 °C for 1 h)	GC-MS	Fish/shellfish and plants	[115]
n.20 from Pax Julia Civitas, Lusitania	FAs, acylglycerols and sterols	Solvent Extraction: CHCl_3_/MeOH (2:1)	BSTFA + 1% TMCS (microwaveoven 700 W for 30 s)	GC-MS	Plant oil	[118]
n.172 from Northwest India	FAs	Solvent Extraction: MeOH/H_2_SO_4_ (4 h at 70 °C)	Direct extraction- derivatization	GC-MS GC-C-IRMS	Animal fat	[4]
n.12 from three sites: Jneneh, Sahab and Tell Abu al-Kharaz.	FAs, alkanols, MAGs, DAGs, sterols	Solvent Extraction: CHCl_3_/MeOH (2:1)	BSTFA + 1% TMCS *	GC-MS	Plant oil and animal fat	[16]
958 potsherds from 14 different sites in Britain	C_16:0_ and C_18:0_	Solvent Extractions: CHCl_3_/MeOH (2:1)	BSTFA + 1% TMCS (70 °C for 1 h) BF_3_-methanol (14% *w*/*v*) (70 °C for 1 h)	GC-MS GC-C-IRMS	Ruminant adipose and dairy fats	[60]
n.63 from Samburu, Kenya	FAs	Solvent Extraction: MeOH/H_2_SO_4_ (1 h at 70 °C)	Direct extraction- derivatization	GC-MS GC-C-IRMS	Ruminant fats	[7]
n.15 from sites in Sardinia and Calabrian n.17 from Sicily	FAs, DAGs, TAGs and estolides	MAE extraction: KOH in ETOH (10% *w*:*v*) 200 W for 60 min	BSTFA + 1% TMCS (60 °C for 30 min)	GC-MS HPLC/ESI-Q-ToF	Cereal	[119]
n.101 from 13 different sites in Japan	FAs and isoprenoid FAs	Solvent Extraction: MeOH/H_2_SO_4_ (4 h at 70 °C)	Direct extraction-derivatization BF_3_-methanol (14% *w*/*v*) (70 °C for 1 h)	GC-MS GC-C-IRMS	Aquatic oils and marine foods	[6]
n. 5 from Sahab, Jordan	FAs	Solvent Extraction: CHCl_3_/MeOH (2:1)	BSTFA + 1% TMCS *	GC-MS	Animal and ruminant fat	[17]
n. 12 from Chrysokamino	FAs	Solvent Extraction: CH_2_Cl_2_/Et_2_O (1:1)	Diazomethane and KOH (25 °C for 24 h)	GC-MS	Plant oil	[120]
n. 10 from Qasr Ibrim, Egypt	TAGs, DAGs, MAGs, FAs, hydroxy FAs and (α, ω)-dicarboxylic acids	Solvent Extraction: CHCl_3_/MeOH (2:1)	BSTFA + 1% TMCS (70 °C for 1 h) BF_3_-methanol (14% *w*/*v*) (75 °C for 1 h)	GC-FID GC-MS GC-C-IRMS	Plant oil	[19]
n.6 from Florencen n.1 from the Pla d’Almatà site (Balaguer, Lleida, Spain)	FAs, MAGs and sterols	Solvent Extraction: CHCl_3_/MeOH (2:1)	BSTFA + 1% TMCS (70 °C for 1 h)	GC-MS	Animal fats, ruminants and vegetable oil	[121]
n. 15 from two sites, one in East Asia and one in Europe (Poland)	FAs and APAAs	Solvent Extraction: MeOH/H_2_SO_4_ (4 h at 70 °C)	BSTFA + 1% TMCS (70 °C for 1 h)	GC-MS GC-C-IRMS	Plant oil	[95]
n.2 from Switzerland	FAs, hydroxy FAs, alkylresorcinols and (α,ω)-dicarboxylic	Solvent Extractions: CH_2_Cl_2_/MeOH (2:1) and MeOH/H_2_SO_4_	BSTFA + 1% TMCS (70 °C for 1 h)	GC-MS	Cereal grains	[122]
n.2 from old quarter of Lekeitio (Basque Country, northern Spain).	FAs, TAGs, (α,ω)-dicarboxylic acid and dihydroxy FAs	MAE extraction: (1) CHCl_3_:Hex (3:2) 600 W for 25 min (2) KOH in ETOH (10%) 200 W for 60 min	BSTFA + 1% TMCS (60 °C for 30 min)	GC-MS HPLC-ESI-Q-ToF	Fish oil	[24]

* Temperature and duration conditions not defined.

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
