# Peer review of "Lipids in Archaeological Pottery: A Review on Their Sampling and Extraction Techniques"

_molecules, 2022, doi:10.3390/molecules27113451_

Round 1
Reviewer 1 Report
This review reports a comprehensive summary and a not very thorough discussion on lipid archaeological biomarkers, and sampling and extraction methodologies. Though the review cites the main papers on the topic, it needs a deep critical revision particularly for the followings:
- the review does not cite and discuss the innovative SFE extraction reported by Deviese (Thibaut Devièse, Alicia Van Ham-Meert, Vincent John Hare, Jasmine Lundy, Peter Hommel, Vladimir Ivanovich Bazaliiskii, and Jayson Orton, Supercritical Fluids for Higher Extraction Yields of Lipids from Archeological Ceramics, Analytical Chemistry 2018 90 (4), 2420-2424, DOI: 10.1021/acs.analchem.7b04913) and Keute (Keute, J., Isaksson, S., Devièse, T., Hein, A., 2021. Insights into ceramic use in prehistoric northwest China obtained from residue analysis: a pilot study on the Andersson Collection at the Museum of Far Eastern Antiquities, Stockholm. Bulletin of the Museum of Far Eastern Antiquities 82, 321-344). This method must be included and discussed.
- Generally, a more detailed discussion on all the extraction methods should be reported giving advantages, disadvantages, reproducibility and reliability of the methods.
- the first paper (and not only reference 72) by Papakosta [Papakosta, V., Smittenberg, R.H., Gibbs, K., Jordan, P., Isaksson, S.,. Extraction and derivatization of absorbed lipid residues from very small and very old samples of ceramic potsherds for molecular analysis by gas chromatography – mass spectrometry (GC – MS ) and single compound stable carbon isotope analysis by gas chromatography. Microchemical Journal 123 (2015) 196–200] where a sulfuric acid and methanol (H2SO4, MeOH 2% v/v) solution is used instead of the traditional chloroform-methanol procedures providing an increased lipid recovery, should be cited and commented.
- In the paragraph “2.2. Free fatty acids as archaeological biomarkers” the discussion on isotopic analysis is poor, and it would benefit by referring to recent papers published by Craig (f.i. Shoda, S., Lucquin, A., Yanshina, O., Kuzmin, Y., Shevkomud, I., Medvedev, V.,Derevianko, E., Lapshina, Z., Craig, O.E., Jordan, P., 2020. Late Glacial hunter-gatherer pottery in the Russian Far East: Indications of diversity in origins and use. Quaternary Science Reviews) and Dunne/Evershed (f.i. Dunne, J., di Lernia, S., ChÅ‚odnicki, M., Kherbouche, F., Evershed, R.P., 2018. Timing and pace of dairying inception and animal husbandry practices across Holocene North Africa. Quaternary International 471, 147–159; Dunne, J., Rebay-Salisbury, K., Salisbury, R.B., Frisch, A., Walton-Doyle, C., Evershed, R.P., 2019. Milk of ruminants in ceramic baby bottles from prehistoric child graves. 189 Nature 574, 246–248).
In the same paragraph, no critical discussion on free fatty acids extracted from wine contained in pottery is reported. In the last years, the presence of wine biomarkers is highly discussed as well as the extraction methods from potteries. This aspect should be also included.
Author Response
Comments and Suggestions for Authors
This review reports a comprehensive summary and a not very thorough discussion on lipid archaeological biomarkers, and sampling and extraction methodologies. Though the review cites the main papers on the topic, it needs a deep critical revision particularly for the followings:
- the review does not cite and discuss the innovative SFE extraction reported by Deviese (Thibaut Devièse, Alicia Van Ham-Meert, Vincent John Hare, Jasmine Lundy, Peter Hommel, Vladimir Ivanovich Bazaliiskii, and Jayson Orton, Supercritical Fluids for Higher Extraction Yields of Lipids from Archeological Ceramics, Analytical Chemistry 2018 90 (4), 2420-2424, DOI: 10.1021/acs.analchem.7b04913) and Keute (Keute, J., Isaksson, S., Devièse, T., Hein, A., 2021. Insights into ceramic use in prehistoric northwest China obtained from residue analysis: a pilot study on the Andersson Collection at the Museum of Far Eastern Antiquities, Stockholm. Bulletin of the Museum of Far Eastern Antiquities 82, 321-344). This method must be included and discussed.
Authors: We thank the Reviewer for the positive comments regarding the review article and we will make our best to improve it based on of his suggestions. As suggested by the Reviewer, the Authors added in the manuscript (page 13 - line 592) the innovative SFE extraction method for lipid analysis in archaeological context. A detailed description of the methodology has been reported highlighting the positive aspects with respect to conventional extraction strategies. As suggested, both the references have been reported in the manuscript.
- Generally, a more detailed discussion on all the extraction methods should be reported giving advantages, disadvantages, reproducibility and reliability of the methods.
Authors: Thanks for the observation. The Authors added the following statement (page 13 - line 614): “In the light of the extraction procedures so far described, the solvent-based extractions are certainly the most widely used in archaeological area due to their reproducibility and reliability. However, these methodologies involve numerous sample preparation steps, sometimes more than necessary, that affect not only the duration of the entire analytical process, but also the quality of the analytical data in term of extraction yield and contamination. With regard to the alternative and innovative strategies, both MAHD and SFE techniques are still not particularly widespread for lipid analyses in archaeological pottery. As a general rule, the most innovative experiments take a long time to be accepted and used instead of the conventional procedures. Nevertheless, the effort required to migrate from a traditional extraction method towards an innovative one is counterbalanced by the fact that equal or higher extraction yield can be obtained in a shorter time. Also, these alternative methodologies include practices according to the miniaturization of the chemical’s consumption, preferential usage of low-toxic reagents and reduction of the waste production.”
- the first paper (and not only reference 72) by Papakosta [Papakosta, V., Smittenberg, R.H., Gibbs, K., Jordan, P., Isaksson, S.,. Extraction and derivatization of absorbed lipid residues from very small and very old samples of ceramic potsherds for molecular analysis by gas chromatography – mass spectrometry (GC – MS) and single compound stable carbon isotope analysis by gas chromatography. Microchemical Journal 123 (2015) 196–200] where a sulfuric acid and methanol (H2SO4, MeOH 2% v/v) solution is used instead of the traditional chloroform-methanol procedures providing an increased lipid recovery, should be cited and commented.
Authors: Thanks for the suggestion. The Authors added the reference Papakosta et al. [116] as suggested by the Reviewer (page 16 - line 685). However, the sample preparation method involving the use of H2SO4, MeOH 2% v/v was already described [Correa-Ascencio and Evershed] and commented in the manuscript.
- In the paragraph “2.2. Free fatty acids as archaeological biomarkers” the discussion on isotopic analysis is poor, and it would benefit by referring to recent papers published by Craig (f.i. Shoda, S., Lucquin, A., Yanshina, O., Kuzmin, Y., Shevkomud, I., Medvedev, V.,Derevianko, E., Lapshina, Z., Craig, O.E., Jordan, P., 2020. Late Glacial hunter-gatherer pottery in the Russian Far East: Indications of diversity in origins and use. Quaternary Science Reviews) and Dunne/Evershed (f.i. Dunne, J., di Lernia, S., ChÅ‚odnicki, M., Kherbouche, F., Evershed, R.P., 2018. Timing and pace of dairying inception and animal husbandry practices across Holocene North Africa. Quaternary International 471, 147–159; Dunne, J., Rebay-Salisbury, K., Salisbury, R.B., Frisch, A., Walton-Doyle, C., Evershed, R.P., 2019. Milk of ruminants in ceramic baby bottles from prehistoric child graves. 189 Nature 574, 246–248).
Authors: Thank for the suggestion. The discussion on isotopic analysis was deepened considering the mentioned references and also other literature papers in the revised version of the manuscript (page 4 – line 175).
In the same paragraph, no critical discussion on free fatty acids extracted from wine contained in pottery is reported. In the last years, the presence of wine biomarkers is highly discussed as well as the extraction methods from potteries. This aspect should be also included.
Authors: Thank for the suggestion. Discussions on papers about free fatty acids extracted from pottery sherds which had contained wine and about wine biomarkers information were added in the revised version of the manuscript (page 4 – line 151).
Reviewer 2 Report
The authors reported reviewed lipids in Archaeological Pottery. The submission can be accepted after revision considering the following points:-
- The quality of the presented data should be improved. The authors can reprint figures from literature with permission to discuss the techniques and experimental setup.
- An in-depth discussion regarding the topic should be added.
- Figure reprinted from the literature should be reprinted with permission. Copyrights should be defined in the Figure’s captions.
- References for lipids should be updated, including these References; Journal of Materials Chemistry B 2014, 2 (42), 7334-7343; TrAC Trends in Analytical Chemistry 2015, 65, 30-46
- The prospective and challenges of each technique should be summarized.
- References should be updated to make the topic broad showing the advantages of the other techniques.
- The language should be revised, and typos should be corrected.
Author Response
Comments and Suggestions for Authors
The quality of the presented data should be improved.
The authors can reprint figures from literature with permission to discuss the techniques and experimental setup.
An in-depth discussion regarding the topic should be added.
Authors: As suggested by the Reviewer. The Authors added the Figure 4 in order to show the experimental set-up of the conventional solvent-based extraction (chloroform/methanol) and direct extraction-derivatization procedure.
Figure reprinted from the literature should be reprinted with permission. Copyrights should be defined in the Figure’s captions.
Authors: We agree with the Reviewer. Permission for the reprint of the Figure 3 was already obtained by the Authors. In fact, the Figure 3’s caption reports as follows: “GC-MS chromatogram of aged gondoic (A) and erucic (B) acid standards. Reproduced with the permission of John Wiley & Sons, Ltd. [91].”
References for lipids should be updated, including these References; Journal of Materials Chemistry B 2014, 2 (42), 7334-7343; TrAC Trends in Analytical Chemistry 2015, 65, 30-46
Authors: reference on lipids has been updated adding (Journal of Materials Chemistry B 2014, 2 (42), 7334-7343) in the revised manuscript (page 12 – line 535). However (TrAC Trends in Analytical Chemistry 2015, 65, 30-46) is not about lipids analysis and therefore it has not been implemented in the text.
The prospective and challenges of each technique should be summarized.
References should be updated to make the topic broad showing the advantages of the other techniques.
Authors: Thanks for the observation. The Authors summarized in the manuscript the prospective and the advantages of each extraction techniques as follows: “In the light of the extraction procedures so far described, the solvent-based extractions are certainly the most widely used in archaeological area due to their reproducibility and reliability. However, these methodologies involve numerous sample preparation steps, sometimes more than necessary, that affect not only the duration of the entire analytical process, but also the quality of the analytical data in term of extraction yield and contamination. With regard to the alternative and innovative strategies, both MAHD and SFE techniques are still not particularly widespread for lipid analyses in archaeological pottery. As a general rule, the most innovative experiments take a long time to be accepted and used instead of the conventional procedures. Nevertheless, the effort required to migrate from a traditional extraction method towards an innovative one is counterbalanced by the fact that equal or higher extraction yield can be obtained in a shorter time. Also, these alternative methodologies include practices according to the miniaturization of the chemical’s consumption, preferential usage of low-toxic reagents and reduction of the waste production.”
The language should be revised, and typos should be corrected.
Authors: Thanks for the observation. The manuscript was revised and the language corrections were made.
Round 2
Reviewer 1 Report
the paper was well improved and can be accepted as it is